# Thermodynamics of the metal-insulator transition in the extended Hubbard model

Malte Schüler[1,2*], Erik G. C. P. van Loon[1,2],
Mikhail I. Katsnelson[3] and Tim O. Wehling[1,2]

**1** Institut für Theoretische Physik, Universität Bremen,
Otto-Hahn-Allee 1, 28359 Bremen, Germany
**2** Bremen Center for Computational Materials Science,
Universität Bremen, Am Fallturm 1a, 28359 Bremen, Germany
**3** Radboud University, Institute for Molecules and Materials,
Heyendaalseweg 135, NL-6525 AJ Nijmegen, The Netherlands

⋆ mschueler@itp.uni-bremen.de

## Abstract

In contrast to the Hubbard model, the extended Hubbard model, which additionally accounts for non-local interactions, lacks systemic studies of thermodynamic properties especially across the metal-insulator transition. Using a variational principle, we perform such a systematic study and describe how non-local interactions screen local correlations differently in the Fermi-liquid and in the insulator. The thermodynamics reveal that non-local interactions are at least in parts responsible for first-order metal-insulator transitions in real materials.



# 1  Introduction

The Hubbard model [1–5] of itinerant electrons with a local interaction is arguably the simplest model describing the competition between kinetic energy and interaction. It is of enormous importance for the understanding of strongly correlated materials and it is thus not surprising that much effort has been spent on unveiling its properties through analytical, numerical, and experimental methods. Since the advent of experiments on cold atoms, which closely simulate the Hubbard model, simulations can even be benchmarked against actual experimental data [6] for high temperatures.

Energetics and entropy are, for instance, well known for a broad parameter regime from cold atom experiments [7, 8] and complementary numerical simulations using the numerical linked cluster expansion (NLCE) [9], dynamical cluster approximation (DCA) [10], determinant quantum Monte Carlo (DQMC) [11], and recently variational cluster approximation (VCA) [12] and cellular dynamical mean field theory [13].

Despite its usefulness for understanding the principles of strongly correlated electron physics, the Hubbard model is quite far from describing real materials: Electrons in real materials interact by long-range Coulomb interaction which in the Hubbard model is approximated by neglecting all but on-site interactions.

A more realistic model Hamiltonian in that sense is the extended Hubbard model that incorporates interactions between electrons on different lattice sites. In contrast to the Hubbard model, the properties of the extended Hubbard model are far less well understood. It is known that strong non-local interactions can induce charge density waves [14–18], influence superconducting properties [19–22], screen local correlations [23–25] lead to band-widening effects [26, 27], first-order metal-insulator transitions [28], and renormalization of Fermi velocities [29, 30]. However, systematic studies of thermodynamic properties are rare and reference data for detailed comparisons and benchmarks between methods especially in context of the metal-insulator transition is missing.

Here, we systematically study the half-filled square lattice in the thermodynamic limit (Sec. 3). The study includes the transition from Fermi-liquid to insulator in the Slater regime (compare Fig. 1a). We describe two distinct mechanisms of how non-local interactions suppress correlation effects: First, in the Fermi-liquid by effectively reducing the local interaction and second, in an insulating state by effectively increasing the hopping amplitude (Sec. 3.2). In the presence of non-local interactions, we find that the competition of these two mechanisms drives a first-order metal-insulator transition which we characterize by its latent heat (Sec. 3.3). We discuss the transition in terms of its relevance in real materials (Sec. 3.4).

We now start by introducing the model and method of approximation in the following Secs. 2.1 and 2.2.

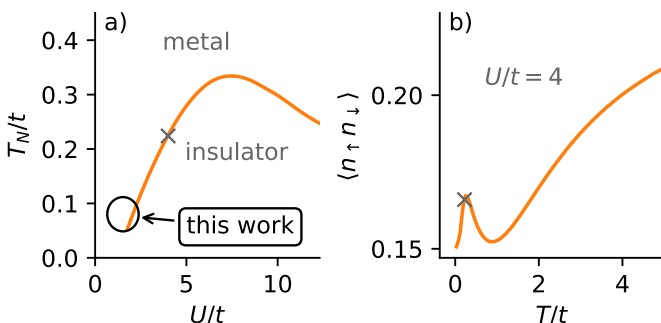

Figure 1: a) Sketch of the Néel temperature in the Hubbard model from CDMFT calculations in Ref. [31]. The regime studied here is highlighted. b) Double occupancy versus temperature in the Slater regime from Ref. [12]. The maximum is close to the Néel temperature (crosses).

## 2 Methods

### 2.1 Extended Hubbard model

The extended Hubbard model with non-local interactions reads,

$$H = -t \sum_{\langle i,j \rangle, \sigma} c_{i\sigma}^{\dagger} c_{j\sigma} + U \sum_i n_{i\uparrow} n_{i\downarrow} + \sum_{i,j} \frac{V_{ij}}{2} n_i n_j, \tag{1}$$

where $c_{i\sigma}^{(\dagger)}$ is the annihilation (creation) operator for electrons on site $i$ and spin $\sigma$, $t$ is the nearest-neighbor hopping amplitude, $U$ is the local interaction, and $V_{ij}$ is the nonlocal interaction between electrons at sites $i$ and $j$. $n_{i\sigma}$ and $n_i$ are the spin-resolved and total occupation operators, respectively. Here, we consider two cases: (a) nearest-neighbor interaction only $V_{01} \equiv V$ and (b) long-range Coulomb interaction $V_{ij} = 1/|\mathbf{r}_i - \mathbf{r}_j|$. For the latter we fix $V_{01}$ by choosing the lattice constant $a = 1/V_{01}$.

### 2.2 Variational principle

We investigate the $U$-$V$-$T$ phase diagram of the extended Hubbard model by approximating its thermodynamic ground state using the Peierls-Feynman-Bogoliubov variational principle [32–34] with a Hubbard model as the effective system [23]. In former applications only the local interaction was varied, capturing the competition between kinetic and potential energy. Here, we extend that scheme by treating both the hopping and local interaction as variational parameters, which captures the competition between energy and entropy. It turns out, that combining both parameters is necessary to properly study the thermodynamics of the first-order transition. For finite systems, we explore the quality of the variational solution for varying only the local interaction, only the hopping, and both in Ref. [27].

The effective Hubbard model with two variational parameters, reads

$$\tilde{H} = -\tilde{t} \sum_{\langle i,j \rangle, \sigma} c_{i\sigma}^{\dagger} c_{j\sigma} + \tilde{U} \sum_i n_{i\uparrow} n_{i\downarrow}. \tag{2}$$

We vary $\tilde{H}$ via the effective hopping parameter $\tilde{t}$ and the effective local interaction $\tilde{U}$ in order to minimize the free energy functional

$$F_v = F_{\tilde{H}} + \langle H - \tilde{H} \rangle_{\tilde{H}}, \tag{3}$$

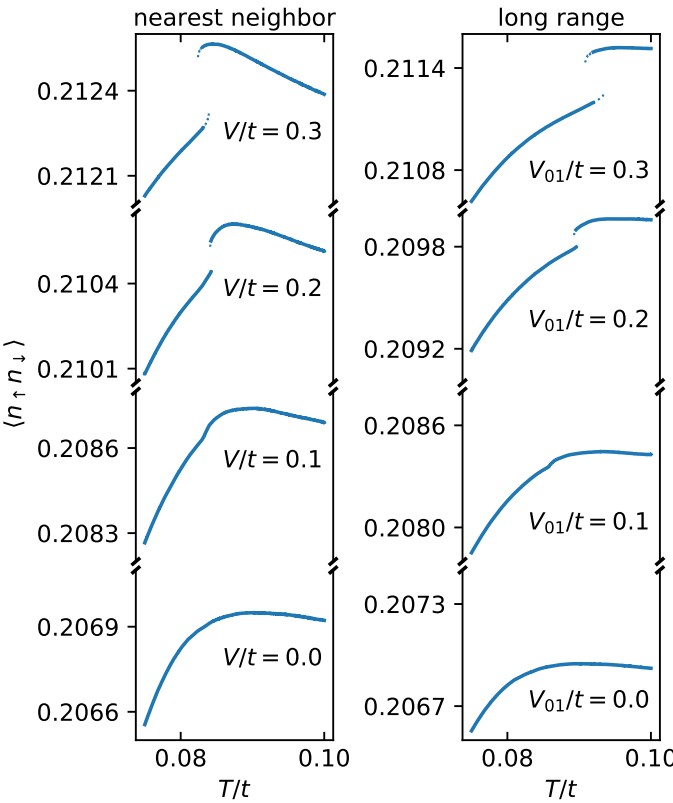

Figure 2: Temperature dependence of the double occupancy for $U/t = 1.9$ and increasing non-local interaction strength (left: nearest-neighbor, right: long range). The dotted lines show coexisting metastable states.

where the expectation value is performed with the density matrix corresponding to $\tilde{H}$. $F_{\tilde{H}}$ is the effective free energy. We perform Determinantal Quantum Monte Carlo (DQMC) [35] simulations for the effective Hubbard model on a $(\tilde{t}, \tilde{U})$ grid, performing finite-size and Trotter discretization extrapolation, as described in Ref. [28]. A two-dimensional Savitzky-Golay filter is used to smooth the data as detailed in Appendix B. We calculate the free energy and entropy by integrating from a non-interacting starting point (see Appendix A). For any value of V, we subsequently evaluate the variational free energy function (3) as a function of $(\tilde{t}, \tilde{U})$ and determine its minimum. Here we use a bivariate spline interpolation to let $\tilde{t}$ and $\tilde{U}$ take continuous values. More details of the computational procedure and error estimates can be found in Appendix B. We provide our raw data and a complete set of input parameters on the Zenodo platform [36].

## 3 Results

### 3.1 Double occupation and Entropy

We start by analyzing how non-local interactions influence observables and thermodynamic properties: Fig. 2 shows the temperature dependence of the double occupancy for different non-local interaction strengths in the cases of nearest-neighbor and long-range Coulomb interaction.

First, we focus on the double occupancy for the Hubbard model ($V = 0$). The temperature

dependence of the double occupancy, shown in Fig. 2, is rather unusual due to a maximum at finite $T$. The maximum exists only in the Slater regime at low temperatures, as previously found in Refs. [12,31] and can be seen more clearly and in the context of its behavior at larger temperatures in Fig. 1b. We briefly review the nature of this maximum to set the stage for the discussion of effects from non-local interactions.

Generally, disregarding long-range anti-ferromagnetic fluctuations (e.g., in DMFT [37] or in frustrated models [38]) the double occupancy first decreases with temperature, since the formation of unordered local moments gains entropy over itinerant electrons (discussed extensively in context of adiabatic or Pomeranchuk cooling in optical lattices [39]). For larger temperatures, the double occupancy eventually increases to meet its non-interacting value in the high-temperature limit. Put together, this gives a minimum close to $T/t = 1$ for parameters investigated here (compare, e.g., Fig. 11 in Ref. [12] for a larger range of temperatures and interaction strengths).

Now, the introduction of long-range anti-ferromagnetic (AF) fluctuations (in more than two dimensions, this corresponds to long-range AF ordering) has a drastic effect on the double occupancy: For small interactions (Slater regime), AF fluctuations decrease the potential energy $UD$ by decreasing the double occupancy $D$. For large interactions (Heisenberg regime) AF fluctuations decrease the kinetic energy through increased hopping probabilities for spins aligned antiparallel [31]. This comes with paying potential energy, i.e., a larger double occupancy $D$. In both regimes, AF fluctuations diminish for larger temperatures (roughly the DMFT Néel temperature), such that we expect $D$ to increase with $T$ in the Slater regime and decrease with $T$ in the Heisenberg regime.

Combining AF fluctuations and local moment formation at small temperatures we find two distinct situations: a) In the Heisenberg regime, both effects cooperate such that we still simply find a minimum in $D(T)$. b) For the Slater regime, AF fluctuations counteract the local moment formation and we find a maximum in $D(T)$ close to the DMFT Néel temperature (compare Fig. 1b) and a minimum at larger temperatures. The case b) is exactly the parameter range we study in this work.

To summarize, two mechanisms are responsible for the maximum at $V = 0$ in Fig. 2. Going to lower temperatures, the double occupancy is lowered to decrease the potential energy (Slater effect). Going to higher temperatures, the double occupancy is lowered to increase the spin entropy (Pomeranchuk effect).

On top of competing formation of local moments and long-ranged AF fluctuations in the Hubbard model, Fig. 2 reveals that non-local interactions introduce multiple effects: First, $V$ generally increases the double occupancy, i.e., decreases local correlation. Second, the position of the maximum is shifted - in the case of nearest-neighbor interaction to smaller temperatures and in the case of long-range interaction to slightly larger temperatures. Third and most strikingly, a large enough $V$ in both cases introduces a discontinuity. On the high temperature side of this discontinuity, the correlation reducing effect of the non-local interactions is stronger, hinting to a more efficient screening where mobile electrons are present. As highlighted by dotted lines, the discontinuity comes with the introduction of meta-stable states. The possibility of hysteresis and the discontinuous behavior of observables indicate a first-order transition, here.

We note that the magnitude of the discontinuity is actually quite small - on the order of 0.0003 for $V/t = 0.3$. This change of about 1.4‰ is larger than the error estimate of below 0.2‰ presented in Fig. 8, such that we rule out numerical artifacts.

In the Hubbard model, the $U$ dependence of the entropy, $S(U)$, is connected to $D(T)$ through a Maxwell relation, $\partial S/\partial U = -\partial D/\partial T$. Thus, the maximum of $S(U)$ visible in Fig. 3 has a common origin with the maximum in $D(T)$. Here, it marks the border between the entropy gain through formation of local moments for small $U$ and the Slater regime where $U$

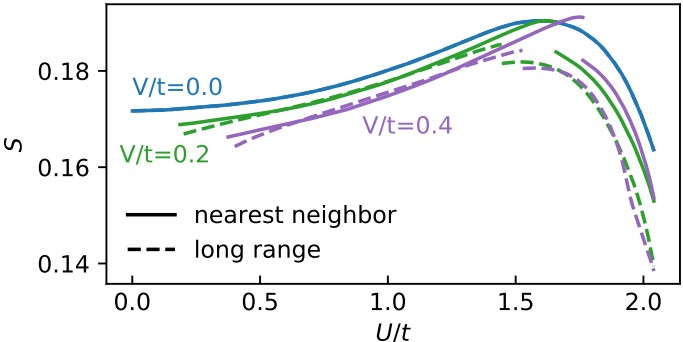

Figure 3: (Color online) $U$ dependence of the entropy per lattice site for $T/t = 0.083$ and different strengths of the nearest-neighbor (solid lines) and long-ranged Coulomb interaction (broken lines). Only data for $U > V$ are shown.

decreases the entropy through AF fluctuations. For even larger $U$, the entropy passes through a minimum to rise again by local moment formation and finally reach $\ln 2$ in the atomic limit.

The introduction of non-local interactions decreases the entropy for most $U$. This decrease is larger on the large-$U$ side of the discontinuity, i.e., in the regime where AF ordering is favored. For the case of nearest-neighbor interactions we observe an increase of the entropy through $V$ for a small regime directly on the low-$U$ side of the discontinuity. For the nearest-neighbor case the maximum is shifted to larger values of $U$.

For the long-range case, the location of the maximum is hardly influenced by non-local interactions for only observing the region of $U$ values above the discontinuity. However, through the introduction of the discontinuity, we find a maximal $S$ at smaller $U$ than in the case of the pure Hubbard model.

Interestingly, in both cases the discontinuity takes place at rather different values of $U$. The discontinuity comes with a larger jump in the case of nearest-neighbor interactions.

Similar to the case of the double occupancy, the relative magnitude of the discontinuity for $V/t = 0.4$ is larger than the error estimate presented in Fig. 8.

## 3.2 Mechanism of screening local correlations.

So far we have gained insight into the effects non-local interactions have on the properties of the Hubbard model by examining the double occupancy and entropy: we have learned that local correlation effects are reduced and under certain circumstances a first-order transition takes place. Now we study the underlying mechanisms of both effects in terms of the free energy of the effective system.

We essentially find three scenarios for how non-local interactions decrease correlations: 1) In the Fermi-liquid, non-local interactions push the system deeper into the Fermi-liquid by decreasing the effective interaction at fixed hopping. 2) Deep in the correlated regime, non-local interactions push the system to effectively lower temperatures by increasing the hopping at fixed interaction strength. 3) In a correlated system close to the metal-insulator transition non-local interactions push the system into the Fermi-liquid regime while increasing the entropy. Under certain circumstances this happens discontinuously. In the following, we will give a detailed analysis of these three regimes, where we from now on focus on the case of nearest-neighbor interaction only, since the case of long-range interaction is qualitatively similar.

Figure 4 shows examples of the variational free energy for all mentioned scenarios. A representative example of the Fermi-liquid regime is shown in Fig. 4 a), where we identify a

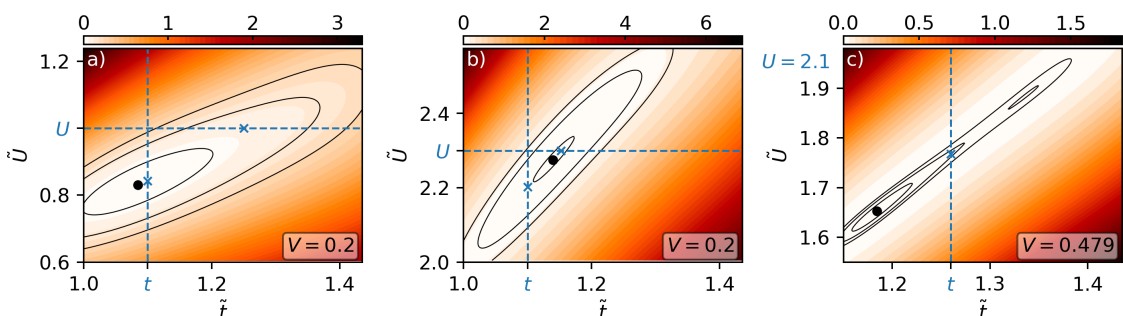

Figure 4: (Color online) Color-coded variational free energy $F_\nu(\tilde{t}, \tilde{U}) \cdot 10^3$ relative to its global minimum for a) a Fermi liquid, b) a strongly correlated regime, and c) at the border between both. The contour lines clarify the low energy structure. Restricted phase space for varying only at fixed $\tilde{t} = t$ and $\tilde{U} = U$ and the respective minima are marked as blue dashed lines and crosses respectively. The temperature is $T = 0.1$.

minimum close to the $\tilde{t} = t$ line. Here, a variation of $\tilde{U}$ alone gives basically the same result as varying both parameters: Correlations are reduced by a quenching of the effective interaction.

In the strongly correlated regime, shown in Fig. 4 b), we find the opposite case: The minimum is located close to the $\tilde{U} = U$ line and the correlations are quenched by increasing the effective hopping.

We observe these two distinct behaviors by tracking how increasing $V$ changes the effective parameters in a $T/\tilde{t}$-$\tilde{U}/\tilde{t}$ phase diagram, shown in Fig. 5. The blue track starts at the square marker in the Fermi regime: Here, only the effective local interaction is changed such that the track moves horizontally to smaller $\tilde{U}/\tilde{t}$ at constant effective temperature $T/\tilde{t}$. The orange track starts in the correlated phase and moves diagonally to smaller $\tilde{U}/\tilde{t}$ *and* smaller effective temperature $T/\tilde{t}$. It does so hardly changing $\tilde{U}$, as shown by the gray dashed lines giving the direction of constant $\tilde{U}$.

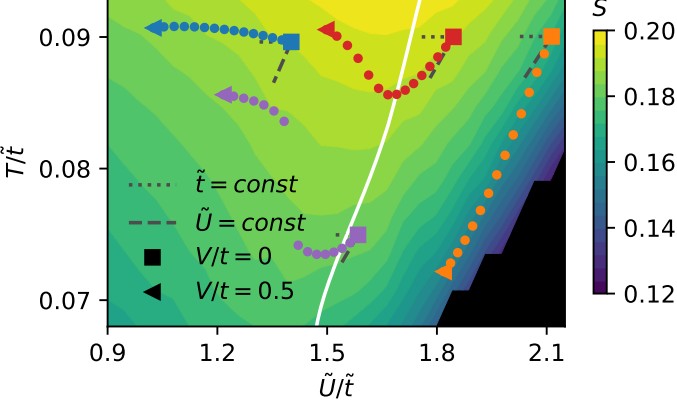

Figure 5: (Color online) Color coded entropy $S$ of the Hubbard model in the region marked by a circle in Fig. 1a). Colored markers represent the effective parameters in this phase space for different nonlocal interactions $V$ (squares: $V = 0$, triangles: $V = 0.5$ and dots in between represent linearly increasing values from $V = 0$ to $V = 0.5$) and $U/t$ ratios (blue: $U = 1.6$, red: $U = 2.1$, yellow: $U = 2.4$ and $t = 1.14$, respectively. Purple: $U = 2.1$ and $t = 1.26$) at $T = 0.1$. The lines show the directions of constant $\tilde{U}$ (dashed) and $\tilde{t}$ (dotted). The white line shows where the variation of $\tilde{t}$ and that of $\tilde{U}$ leads to the same free energy gain for infinitesimal $V$ (see text).

We identify the different regimes by expanding the free energy functional (Eq. 3) parabolically in $V$. One can then explicitly show whether the variation of $\tilde{t}$ or that of $\tilde{U}$ leads to a smaller free energy for infinitesimal $V$ as shown in Ref. [27]. One finds that the variation of $\tilde{t}$ minimizes the free energy if

$$\frac{1}{2}\frac{(\partial_{\tilde{t}}\langle n_0 n_1\rangle)^2}{\partial_{\tilde{t}}\langle c_1^\dagger c_0\rangle} > z\frac{\left(\partial_{\tilde{U}}\langle n_0 n_1\rangle\right)^2}{\partial_{\tilde{U}}\langle n_{0\uparrow}n_{0\downarrow}\rangle}, \tag{4}$$

and vice versa, where $z$ is the number of nearest neighbors. The solid white line in Fig. 5 shows where the left and right expressions coincide. Left to the line, the variation of $\tilde{U}$ minimizes the free energy for infinitesimal $V$, right to the line that of $\tilde{t}$ does so. The line basically follows the maxima of $S(U)$ visible as dips in the contour lines.

We finally come to the last and most interesting case of non-local interactions inducing a (discontinuous) metal-insulator transition. We show a representative free energy surface in Fig. 4 c), where we find two local minima. Interestingly, both minima are found with similar $\tilde{U}/\tilde{t}$ ratio such that the competing configurations differ only in effective temperature $T/\tilde{t}$. Here the global minimum is realized for the higher effective temperature. By decreasing the non-local interaction $V$ slightly the free energy landscape changes such that the minimum at low temperature (high $\tilde{t}$) is the global minimum: non-local interactions induce a first-order phase transition.

We also investigate this third scenario in terms of how the effective parameters in the $T/\tilde{t}$-$\tilde{U}/\tilde{t}$ phase diagram behave. Both the red and magenta track start for $V=0$ at the square markers in the correlated regime such that non-local interactions lead to an increase of the hopping. In the first case we find a smooth transition to a Fermi-liquid. In the second case, we find a critical $V$ for which the track jumps at constant $\tilde{U}/\tilde{t}$ to larger effective temperature. The entropy, which is color coded in the same figure, increases during the jump with the effective temperature.

This general description of how non-local interactions screen local correlation effects extends findings from GW+DMFT in Ref. [24] which show that the partially (only by non-local interactions) screened local interaction at zero frequency strongly depends on $V$ in the Fermi-liquid regime, while it is nearly constant in the Mott-insulator. However, in contrast to our findings the transition from strong to weak screening happens continuously in GW+DMFT.

### 3.3 Critical non-local interaction

In the following, we identify the critical non-local interaction $V_c$, defined as the smallest $V$ for which a discontinuous transition takes place. We present the $U$-dependence of $V_c$ and the accompanying jump in entropy $\Delta S$ (which is proportional to the latent heat) for $t=1.26$ and $T=0.1$ in Fig. 6.

At this temperature, the smallest $V_c(U)$ is $\sim 0.1t$, which is significantly smaller than the strong coupling estimate for the charge density wave phase ($V \gtrsim U/4$). This rules out any kind of masking by a charge-density-wave phase. The jump in entropy is on the order of $10^{-3}$, on which we base the assessment of observability in experimental setups in the next section. Importantly, finding $V_c(U) < U/4$ also leads to the possibility of simulating the models containing the prospective discontinuity by numerically exact simulations, e.g., bond-centered auxiliary field Monte Carlo [40], since $V < 4U$ is the parameter regime, where no sign problem occurs in the square lattice.

We note that for fewer variational degrees of freedom (by fixing either $\tilde{t}=t$ or $\tilde{U}=U$) we do not find a first-order phase transition for the parameters shown in Fig. 6: The increased variational freedom is crucial, here, to properly describe the competition between minimizing

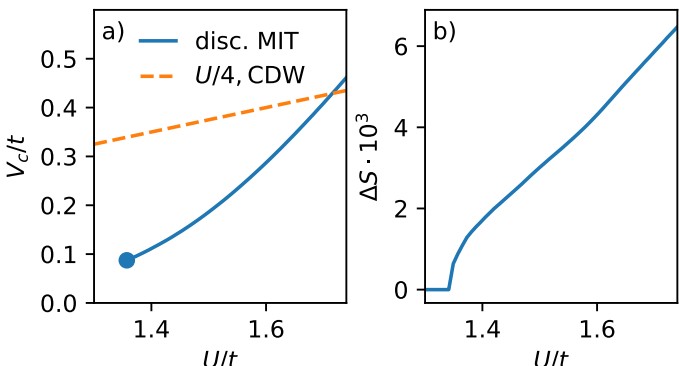

Figure 6: (Color online) a) Critical nonlocal interaction $V_c$ for which a first-order transition takes place. The dot marks the smallest $U/t$ at which a first-order transition takes place. b) Change of the entropy due to the phase transition. The temperature is $T/t = 0.079$ for a) and b).

the potential energy (via $\tilde{U}$) in the Fermi liquid and minimizing the kinetic energy (via $\tilde{t}$) in the Mott insulator.

### 3.4 Comparison to experimentally observed first-order transitions

Experimentally observed discontinuous metal-insulator transitions can be classified into two groups: The first involves transitions where electronic and lattice degrees of freedom participate, e.g., in the case of the metal-insulator transition in $VO_2$. These typically come with a large change of entropy ($\Delta S \sim 1.2$ per V atom).

The second group involves purely electronic transitions with a typically smaller jump in entropy. Two examples of the second group are the quantum magnet TiOCl with an entropy change smaller than $\Delta S = 0.1$ per unit cell [41] and $Sm_2IrIn_8$ where an antiferromagnetic transition leads to $\Delta S = 0.08$ per unit cell. [42] Even smaller jumps are observed for an electric-field induced transition in the ferroelectric ceramic $Ba_{0.73}Sr_{0.27}TiO_3$ with $\Delta S = 0.01$ per Ti atom [43] and for the metal-insulator transition in $NiS_{2-x}Se_x$ with $\Delta S$ between 0.003 and 0.08 per unit cell. [44, 45]

Since we do not consider any lattice degrees of freedom, we should compare the jumps in entropy found here to the second group. The largest jumps found (which are not masked by a charge density wave) are on the order of $\Delta S \sim 0.005$, which is well inside the range associated with purely electronic transitions in real materials.

A further interesting group of materials showing first-order metal-insulator transitions are alkali-doped fullerides [46] and layered organic superconductors [47, 48] since they both host considerable non-local interactions, as found in ab-initio calculations [49, 50]

## 4 Conclusion

We have investigated thermodynamic properties of the metal-insulator transition in the extended Hubbard model, established a reference for future numerical studies, and found discontinuous transitions accompanied with latent heat which is in the range of what is experimentally observed in real materials whenever mainly electronic transitions are involved.

Generally, we have found distinct mechanisms for the suppression of correlation effects in the Fermi-liquid and Mott insulating regime: in the metallic phase the influence of non-local interactions is effectively described by a reduction of the local interaction $U$. Conversely, in the

insulating phase it is described by an increase of the hopping ($t$). The discontinuous transition finally is described by simultaneously decreasing $t$ and $U$ at a fixed ratio, which comes with an increase of the effective temperature and therefore the entropy.

The regime of parameters for which we find discontinuous Mott transitions rules out masking by charge density waves.

By comparing jumps of the entropy to experimentally observed first-order transitions, we are confident that the mechanism for a first-order transition is of experimental relevance and that it should be a significant contributor to discontinuous metal-insulator transitions in real materials, particularly in situations where lattice distortions are not relevant.

# Acknowledgements

The authors acknowledge helpful discussions with Erik Koch, Alloul Henri, and Alexander Lichtenstein.

**Funding information** MS, EGCPL, and TOW acknowledge financial support of DFG via RTG QM$^3$. MIK acknowledges financial support of NWO via Spinoza Prize. The authors acknowledge the North-German Supercomputing Alliance (HLRN) for providing computing resources via project number hbp00046 that have contributed to the research results reported in this paper.

# A    Calculation of the free energy and entropy

In the DQMC method, we have no direct access to the free energy. However, we do have access to its first derivative the double occupancy:

$$\frac{\partial F_{\tilde{H}}}{\partial \tilde{U}} = \langle n_\uparrow n_\downarrow \rangle = D. \tag{5}$$

We obtain the free energy at any $\tilde{U}$ by integrating from $\tilde{U}_0 = 0$,

$$F_{\tilde{H}}(\tilde{t}, \tilde{U}) = \int_{\tilde{U}_0}^{\tilde{U}} N D(\tilde{t}, \tilde{U}') d\tilde{U}' + F_{\tilde{H}}(\tilde{t}, \tilde{U}_0), \tag{6}$$

where the free energy in the analytically solvable noninteracting case is given by

$$F_{\tilde{H}}(\tilde{t}, \tilde{U}_0 = 0) = -\frac{2}{\beta N_k} \sum_k \log(1 + \exp(-\beta(\tilde{\varepsilon}_k - \mu))), \tag{7}$$

where $\tilde{\varepsilon}_k$ are the single-particle energies of the non-interacting system with hopping amplitude $\tilde{t}$. We calculate the integral over the double occupancy numerically using the composite trapezoidal rule.

We approximate the original system's entropy $S = (\langle H \rangle - F_H)/T$ using Eq. 3 and evaluating the expectation value with the effective density matrix for optimized $\tilde{U}$ and $\tilde{t}$:

$$S \approx (\langle H \rangle_{\tilde{H}} - F_\nu)/T = (\langle \tilde{H} \rangle_{\tilde{H}} - F_{\tilde{H}})/T = \tilde{S}. \tag{8}$$

In a recent work, Walsh et al. [13] extract the entropy of the Hubbard model from the Gibbs-Duhem relation. Their approach is based on an integration over chemical potential, that is, it requires simulations away from half-filling. In contrast, the coupling-constant integration presented here can be performed using only half-filled simulations. For our purpose here, that is a substantial advantage since DQMC suffers from a sign problem away from half-filling.

## B   Error estimation

Performing an error estimate for the results of the variational principle is difficult due to a multitude of error sources: a) statistical (finite sampling) and systematic (finite size and finite Trotter discretization) DQMC errors, b) uncertainties in the evaluation of the free energy, and c) errors from the limits of the variational principle.

   In Secs. B.1 and B.2 we explain how we deal with the statistical and systematic DQMC errors. In the remaining sections B.3 and B.4 we discuss errors in the free energy and convergence of sums of charge correlation functions over neighbors throughout the lattice. The most difficult error to assess is the intrinsic error introduced by the limitations of the variational principle. We have performed benchmarks in Ref. [25] for the case of only varying $\tilde{U}$ and we expect qualitatively similar results here. I.e., that the variational principle performs naturally well in describing properties concerning spin fluctuations (e.g., the Mott transition) but bad at properties concerning charge fluctuations (e.g., the charge density wave).

### B.1   Statistical DQMC Errors

We use the determinant quantum Monte Carlo method (DQMC) [35] implemented in the QUEST code [1]. We solve Hubbard models in steps of $\Delta\tilde{t} = 0.02$ and $\Delta\tilde{U} = 0.1$ and use the fact that in the case of a finite system all observables behave smoothly with $\tilde{U}$ and $\tilde{t}$. We drastically reduce the statistical error by using a Savitzky-Golay approach [51]: We fit a two-dimensional polynomial,

$$g(\tilde{t}, \tilde{U}) = \sum_{nm}^{NM} a_{nm} \tilde{t}^n \tilde{U}^m,  \tag{9}$$

to the data in a box of widths $w_{\tilde{t}}$ and $w_{\tilde{U}}$ around the point $(\tilde{t}_0, \tilde{U}_0)$. We use $w_{\tilde{t}} = 1.0$, $w_{\tilde{U}} = 1.0$, and third order polynomials. We additionally give larger weight to data close to $(\tilde{t}_0, \tilde{U}_0)$ as to those further away by using a tricubic weighting function $\left(1 - d^3\right)^3$ where the distance is defined as $d = \max\{|\tilde{t} - \tilde{t}_0|/w_{\tilde{t}}; |\tilde{U} - \tilde{U}_0|/w_{\tilde{U}}\}$. For all parameters we have performed 500 warm up sweeps and 50000 measurement sweeps.

### B.2   Systematic DQMC errors

Next to statistical errors stemming from finite sampling times in the Monte Carlo procedure which we deal with by smoothing (c.f. Sec. B.1), all observables acquire a systematic error by the Trotter-decoupling for finite imaginary time differences $\Delta\tau$. In many cases one achieves *qualitatively* correct results by choosing small enough values of $\Delta\tau$ (e.g., by choosing $\Delta\tau \sim \sqrt{0.125/U}$, as suggested in Ref. [52]). However, in the case of calculating the free energy (and the entropy) by integrating the double occupancy over $\tilde{U}$ we magnify the systematic error (while averaging out the statistical error), which leads to even qualitatively wrong results.

   We show this for simulations [53] for a larger set of $\tilde{U}/\tilde{t}$. In Fig. 7 we show the entropy calculated from observables for finite $\Delta\tau$ and from those which are extrapolated to $\Delta\tau = 0$. The entropy for the case of finite $\Delta\tau$ is negative for large $U$ which breaks basic thermodynamic laws. This is the case, even though both values of $\Delta\tau$ shown here are smaller than required by the rule of thumb given above for all values of $\tilde{U}$. Interestingly, this effect worsens for smaller temperatures. Only for the data extrapolated to $\Delta\tau = 0$, do we find strictly positive entropy. In the end, we alleviate finite-size and Trotter errors by extrapolating from finite

---

[1] "QUantum Electron Simulation Toolbox" QUEST 1.4.9 A. Tomas, C-C. Chang, Z-J. Bai, and R. Scalettar, (http://quest.ucdavis.edu/)

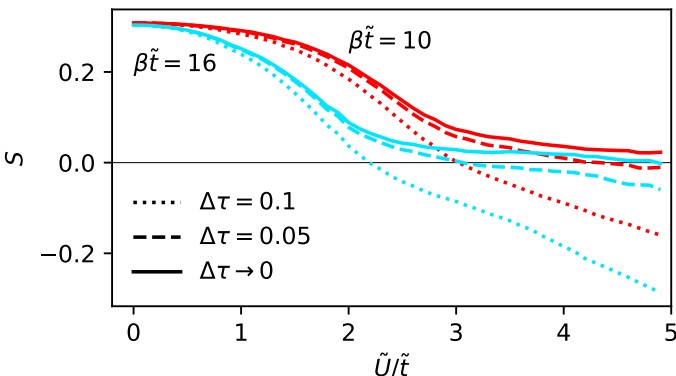

Figure 7: (Color online) Entropy from DQMC simulations with $\Delta\tau = 0.1$ (dotted lines), $\Delta\tau = 0.05$ (dashed lines) and from extrapolated observables for $\beta\tilde{t} = 10$ (red) and $\beta\tilde{t} = 16$ (blue). The linear size of the simulated square lattice is $L = 8$.

Trotter discretizations of $\Delta\tau = 0.2$, 0.1, and 0.05 and linear lattice sizes of $L = 8$, 10, and 12 for the square lattice following Refs. [54] and [28].

Determinantal Quantum Monte Carlo maps the Hubbard model at finite temperature to an auxiliary $d+1$ dimensional Ising model. In this transformation, the temperature is discretized and Suzuki-Trotter discretization errors appear. The observation of negative entropy at $\Delta\tau > 0$ in Fig. 7 shows that the discretized system is not thermodynamic, in the sense that basic laws of thermodynamics are violated. Only after the extrapolation to $\Delta\tau = 0$ are they restored. Validating these thermodynamic laws serves as a useful check on the extrapolation process and the quality of the Monte Carlo data.

The free energy also provides a useful validation of the Monte Carlo data. The free energy at constant temperature is a function of $\tilde{t}$ and $\tilde{U}$. The nearest-neighbor Green's function and the double occupancy are the observables conjugate to these parameters, i.e., they could be obtained as first derivatives of the free energy. (In practice, we actually obtain the observables from DQMC and then get the free energy by integration.) Thus, the derivative of the double occupancy w.r.t. $\tilde{t}$ is a second derivative of the free energy and by Clairaut's theorem it is equal to the derivative of the Green's function w.r.t. $\tilde{U}$, modulo prefactors from spin factors. Furthermore, the free energy is a state function and integrating the free energy from $t_1$, $U_1$ to $t_2$, $U_2$ should be independent of the path taken through the $t-U$ plane. These equalities hold for the exact solution of the Hubbard model, for Monte Carlo data it is subject to statistical errors and systematic $\Delta\tau$ errors and provides a useful estimate of the quality of the data.

### B.3 Errors related to the free energy

To determine the parameters $\tilde{U}_0$ and $\tilde{t}_0$ which minimize the free energy functional $F_\nu$ defined in Eq. 3, we calculate the functional for a fixed mesh of $\tilde{U}$ and $\tilde{t}$ and perform a minimization using interpolating splines of third order. The accuracy of this approach crucially depends on the accuracy of the observables entering $F_\nu$, i.e., the double occupancy, nearest-neighbor correlation function and Green's function, and the free energy (from Eq. 6).

We assess the accuracy of the functional by checking if its minimum is located at $\tilde{U} = U$ and $\tilde{t} = t$ for the trivial case of $\tilde{H} = H$, (no non-local interactions, $V = 0$). In the upper panels of Fig. 8 we show the deviations from the expected result in terms of the variational parameters, i.e., $\Delta U = \tilde{U}_0 - U$ and $\Delta t = \tilde{t}_0 - t$. Both values show considerable deviations from the ideal behavior ($\Delta U = \Delta t = 0$), which worsens for larger $U$. The errors are strongly correlated with each other, i.e., we find large positive $\Delta U$ where we do so for $\Delta t$. The jumps in $\tilde{t}$ corresponding to the first order transition, which can be seen in Fig. 5, are about 0.15 -

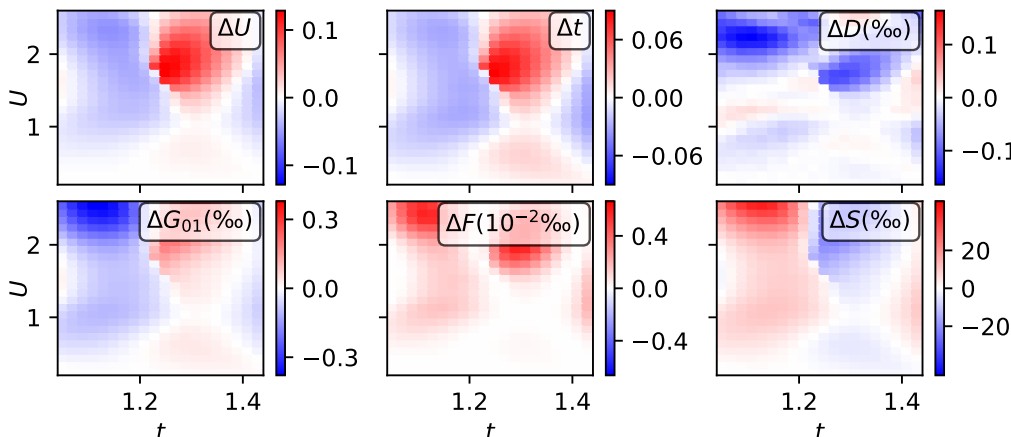

Figure 8: (Color online) Error in finding the correct minimum of the free energy for $V = 0$.

twice the size of the error estimate shown here.

Since $\tilde{U}$ and $\tilde{t}$ are only variational parameters without any direct physical meaning, it is more interesting to investigate the relative deviations of observables defined as $\Delta O = (\tilde{O} - O)/\tilde{O}$. We find the relative errors of the double occupancy and nearest-neighbor Green's function to be below one per mille. The error in the free energy is even two orders of magnitude smaller. This roots first in a generally quite flat free energy surface and second - since the errors in $\tilde{U}$ and $\tilde{t}$ are positively correlated - in the weak dependence of the free energy in constant $\tilde{U}/\tilde{t}$ direction (c.f. surfaces shown in Fig. 4).

Since the entropy does not only depend on $\tilde{U}/\tilde{t}$ but also strongly on $\tilde{t}$ via the effective increase of the temperature $T/\tilde{t}$, we find more considerable errors, here. The deviations of the entropy from the expected result for the parameters studied in this work are between 1% and 2%.

## B.4 Finite-size convergence for the Coulomb case

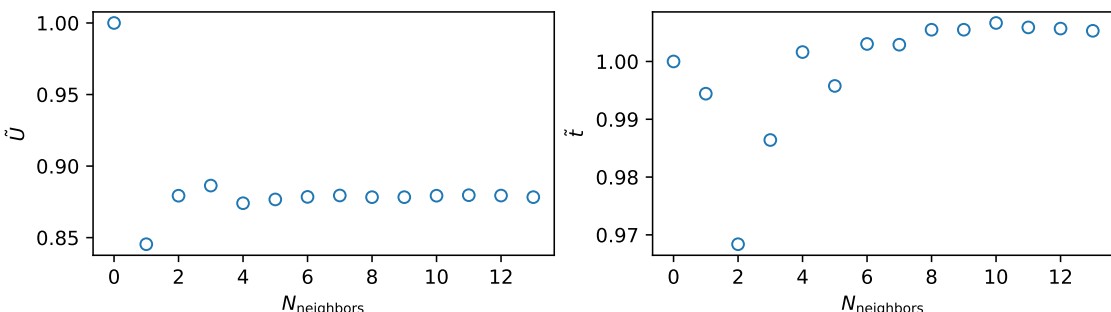

Figure 9: Convergence of the effective parameters with the number of neighbors considered in their calculation in the Coulomb case for $U = 1$, $t = 1$, $V_{01} = 0.2$, and $\beta = 10$.

For the case of long-range Coulomb interaction, the calculation of the free energy function (Eq. 3), involves a sum over neighbors: $\sum_i V_{0j} \langle n_0 n_j \rangle$. Although we perform a finite size extrapolation of the observables $\langle n_0 n_j \rangle$, we can only do this for observables which are actually present in the smallest lattice considered (which are 14 non-equivalent neighbors in the case of an $8 \times 8$ lattice). We therefore have to check how the sum and thereby the effective

parameters converge. We plot the convergence of $\tilde{t}$ and $\tilde{U}$ in Fig. 9. Both basically converge for 8 neighbors, such that finite size errors in the case of long-range Coulomb interaction are negligible. The fast convergence relates to the combination of the $1/r$ behavior of $V(r)$, the exponential decay of spatial correlations for finite $\tilde{U}$ and a checkerboard like oscillation of the sign of $\partial/\partial\tilde{U}\,\langle n_0 n_j\rangle$ at small $\tilde{U}$ [55].

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
