# Peer review of "Thermodynamics of the metal-insulator transition in the extended Hubbard model"

_SciPost Physics, doi:SciPost Phys. 6, 067 (2019)_

## Round 2 · Referee Report · Anonymous (Referee 1) · 2019-5-4

Strengths

1 - An interesting study on thermodynamics of the metal-insulator transition in the extended Hubbard model.
2 - Comprehensive numerical calculation that clarifies how non-local interactions screen local correlations differently in the Fermi-liquid and in the insulator.
3 – Numerical indications that non-local interactions are at least in parts responsible for first-order metal-insulator transitions in real materials.

Weaknesses

1- Numerically very subtle calculation.

Report

This work takes a variational approach to study the extended Hubbard model having interactions between electrons on different lattice sites. In contrast with the on-site Hubbard model, the properties of the extended Hubbard model are far less well understood. Especially, as is emphasized by the authors, systematic studies of thermodynamic properties are rare and reference data for detailed comparisons and benchmarks between methods is missing.
In the phase diagram, there is a phase transition from Fermi-liquid to insulator. The authors focus on the so-called Slater regime with small interactions. There are two mechanisms how non-local interactions suppress correlation effects. Namely, the non-local interactions effectively reduce (enhance) the local interaction (transfer hopping) in the Fermi liquid (insulator). They found that the competition of these two mechanisms drives a first-order metal-insulator transition.
The authors perform a comprehensive numerical study and present a variety of plots which corroborate their claim. This work will stimulate further interest in the extended Hubbard model and I would like to recommend publication in SciPost Physics.

Requested changes

It would be better to explicitly mention that the change in the double occupancy plotted in Fig. 2 is very small. Even in the case of V/t=0.3, the jump is only 0.0003. The energy scale of the structure plotted in Fig.4 is also 0.0001. I understand that they performed the calculation very carefully, and the variational method which they employed worked quite successfully in their previous works. However, their approach is not the unique way to map the extended Hubbard model onto the Hubbard model. We may think about the possibility where {\tilde U} in the effective model has \omega dependence. I wonder how the present result is modified when we improve the mapping from the extended Hubbard model to the on-site Hubbard model.

---

## Round 2 · Referee Report · Anonymous (Referee 2) · 2019-5-12

Strengths

In this manuscript the authors study thermodynamic properties of the extended Hubbard model in 2D square lattice. This analysis is potentially important for further progress in understanding the extended Hubbard model, as there are no systematic analysis for such model yet.
The authors use the varioational method and map the original extended Hubbard model to the effective Hubbard model which they can solve using DQMC.
One of the main findings of this paper is that the authors claim that there is a first order metal-insulator transition induced by non-local intersite interactions V, observed as jumps in the double-occupancy and entropy analysis.

Weaknesses

The major claim of this paper is based on the very small jumps in the double occupancy and entropy~10^-4. It is really not clear whether it is a numerical artifact, the noise, or the artifact of the particular variational method employed in this paper.
The authors state that they do not observe such first order transition in variational method is used only for the U term (with no hopping renormalization). Hence, it is not clear how valid this varioational approach is, some benchmarking is really necessary.
For this reason I am reluctant to recommend this paper for publication.

Report

I am reluctant to recommend this paper for publication.

Requested changes

  1. The authors should say explicitly how the entropy calculations was done.
  2. In Fig. 4 and Fig. 5, the data are presented in $t_tilde$ and $U_tilde$ paramter space. It is really not clear how these numbers are related to real U and t values.
  3. The authors claim that they find the band broadening effect of non-local interactions V. Their claim is based on some speculations of free energy behavior in t_tilde parameter space. If the authors mean they see a similar effect as in REf.https://journals.aps.org/prb/abstract/10.1103/PhysRevB.95.245130, they should perhaps find a better way to support their claim.
  4. Why in eq. 7, the original lattice dispersion function $\epsilon_k$ and not for the effective Hubbard model $\tilde{\epsilon_k}$ is used.
  5. Why the energy units (hopping $t$) are changes through the paper? In Fig. 6 the authors used $t=1.26$ and in Fig. 9, $t=1$ is used. This way your are studying the original extended Hubbard model for different bandwidth values.

---

## Round 3 · Author Response

We thank the referees for their careful reading of the manuscript and for their useful questions.
Based on the comments we have carefully revised our manuscript to make it clearer and more convincing. Both referees remarked that the magnitude of our findings on first-order transitions are warningly small. We have addressed this issue by adding a discussion on the significance of our findings in relation to the error estimates which have both numerical and methodological sources.
Here we provide point-by-point answers to all referees comments.
Referee 1:
\emph{1) It would be better to explicitly mention that the change in the double occupancy plotted in Fig. 2 is very small. Even in the case of $V/t=0.3$, the jump is only 0.0003. The energy scale of the structure plotted in Fig.4 is also 0.0001.}
Following also the critique of the second referee, we now highlight the magnitude of the changes in the text and have added a discussion on the significance of the magnitude of the discontinuity with respect to the error estimates shown in the appendix.
\emph{I understand that they performed the calculation very carefully, and the variational method which they employed worked quite successfully in their previous works. However, their approach is not the unique way to map the extended Hubbard model onto the Hubbard model. We may think about the possibility where $\tilde U$ in the effective model has $\omega$ dependence. I wonder how the present result is modified when we improve the mapping from the extended Hubbard model to the on-site Hubbard model.}
The extension of the effective model by a frequency dependent $\tilde U (\omega)$ is an interesting approach to improve the mapping. Unfortunately this is out of reach in our current setup and probably worth a separate publication. Our method crucially depends on the possibility to be able to solve the effective model. This is only true for the Hubbard model with static interaction, by using the DQMC method.
Typically, mappings of an extended Hubbard model to a model with frequency dependent local interaction come with severe approximations which trace back to neglecting nonlocal correlations, like in extended dynamical mean-field theory (EDMFT). Diagrammatic extensions to EDMFT, which reintroduce these nonlocal correlations (like GW+EDMFT or the dual boson approach) are typically themselves approximative and computationally demanding.
Referee 2:
\emph{The major claim of this paper is based on the very small jumps in the double occupancy and entropy $\sim 10^{-4}$. It is really not clear whether it is a numerical artifact, the noise, or the artifact of the particular variational method employed in this paper.}
As also pointed out by the other referee the discontinuities in, e.g., the double occupancy are rather small. They are, however, significantly larger than the error estimates presented in figure 8 in the appendix. We have added corresponding discussions to the manuscript.
\emph{The authors state that they do not observe such first order transition in variational method is used only for the U term (with no hopping renormalization). Hence, it is not clear how valid this varational approach is, some benchmarking is really necessary.}
Unfortunately, we cannot follow the referee's line of argumentation here: How does not finding a feature in an inferior version of a method weaken the validity of a finding in the superior version? Our statement that lower variational degrees of freedom do not show a first-order transition is supposed to highlight that the physical mechanism behind this transition is the competition of kinetic and potential energy minimization, described in our variational approach by an abrupt change from a regime where mostly the effective interaction changes to one where mostly the effective hopping changes with $V$.
We have added a short discussion to the corresponding paragraph, clarifying the physical reason of why two parameter are crucial.
Related to this comment, we have showed in an earlier work (Phys. Rev. B 97, 165135 (2018)), that the variational principle with only one variational parameter does give a first-order transition, but this happens at much larger values of the non-local interaction $V$, where charge-density waves become relevant.
Concerning benchmarking of the variational method, we refer to our publication (Phys. Rev. B 94, 165141 (2016)) where we have performed promising benchmarks against dual boson calculations for the case of a single variational parameter on the square lattice. Additionally, our recent preprint (https://arxiv.org/abs/1901.11257) shows benchmarks of the method for the case of two variational parameters for exactly solvable systems.
\emph{1) The authors should say explicitly how the entropy calculations was done. }
We have included details of how we calculate the entropy to Appendix A on the free energy calculation.
\emph{2) In Fig. 4 and Fig. 5, the data are presented in $\tilde t$ and $\tilde U$ parameter space. It is really not clear how these numbers are related to real $U$ and $t$ values.}
In Fig. 4, the parameters of the original model (the real parameters $U$, $t$) are highlighted as blue lines in the space of variational parameters. This directly shows the influence nonlocal interactions have on the variational parameters. In Fig. 5, the parameters of the original model are those corresponding to the square markers, since $V=0$ means $\tilde U = U$ and $\tilde t = t$. For clarity, we have added the parameters of the original model in the caption of Fig. 5.
\emph{3) The authors claim that they find the band broadening effect of non-local interactions $V$. Their claim is based on some speculations of free energy behavior in $\tilde t$ parameter space. If the authors mean they see a similar effect as in Ref.~\url{https://journals.aps.org/prb/abstract/10.1103/PhysRevB.95.245130}, they should perhaps find a better way to support their claim.}
We actually do not mention band-widening in our manuscript. We did so on purpose although it may seem tempting to interpret a renormalization of the effective parameter $\tilde t$ as a band renormalization. First, however, $\tilde t$ only determines the band-width in the non-interacting (and weakly interacting) case. Second, the variational principle we use leads to variational parameters which optimize the free energy, i.e., ground-state properties, and not spectral properties.
\emph{4) Why in eq. 7, the original lattice dispersion function ϵk and not for the effective Hubbard model $\tilde \varepsilon_k$ is used.}
Eq. 7 of course contains the dispersion of the effective model. We have fixed our ambiguous notation in the revised manuscript.
\emph{5) Why the energy units (hopping t) are changes through the paper? In Fig. 6 the authors used $t=1.26$ and in Fig. 9, $t=1$ is used. This way your are studying the original extended Hubbard model for different bandwidth values.}
In our work, we decided to fix the energy units by the temperature ($T=0.1$) rather than a fixed hopping. Using the variational principle with the hopping $\tilde t$ as an effective parameter, it is quite natural to also consider a variable original hopping $t$.
Based on the comments we have carefully revised our manuscript to make it clearer and more convincing. Both referees remarked that the magnitude of our findings on first-order transitions are warningly small. We have addressed this issue by adding a discussion on the significance of our findings in relation to the error estimates which have both numerical and methodological sources.
Here we provide point-by-point answers to all referees comments.
Referee 1:
\emph{1) It would be better to explicitly mention that the change in the double occupancy plotted in Fig. 2 is very small. Even in the case of $V/t=0.3$, the jump is only 0.0003. The energy scale of the structure plotted in Fig.4 is also 0.0001.}
Following also the critique of the second referee, we now highlight the magnitude of the changes in the text and have added a discussion on the significance of the magnitude of the discontinuity with respect to the error estimates shown in the appendix.
\emph{I understand that they performed the calculation very carefully, and the variational method which they employed worked quite successfully in their previous works. However, their approach is not the unique way to map the extended Hubbard model onto the Hubbard model. We may think about the possibility where $\tilde U$ in the effective model has $\omega$ dependence. I wonder how the present result is modified when we improve the mapping from the extended Hubbard model to the on-site Hubbard model.}
The extension of the effective model by a frequency dependent $\tilde U (\omega)$ is an interesting approach to improve the mapping. Unfortunately this is out of reach in our current setup and probably worth a separate publication. Our method crucially depends on the possibility to be able to solve the effective model. This is only true for the Hubbard model with static interaction, by using the DQMC method.
Typically, mappings of an extended Hubbard model to a model with frequency dependent local interaction come with severe approximations which trace back to neglecting nonlocal correlations, like in extended dynamical mean-field theory (EDMFT). Diagrammatic extensions to EDMFT, which reintroduce these nonlocal correlations (like GW+EDMFT or the dual boson approach) are typically themselves approximative and computationally demanding.
Referee 2:
\emph{The major claim of this paper is based on the very small jumps in the double occupancy and entropy $\sim 10^{-4}$. It is really not clear whether it is a numerical artifact, the noise, or the artifact of the particular variational method employed in this paper.}
As also pointed out by the other referee the discontinuities in, e.g., the double occupancy are rather small. They are, however, significantly larger than the error estimates presented in figure 8 in the appendix. We have added corresponding discussions to the manuscript.
\emph{The authors state that they do not observe such first order transition in variational method is used only for the U term (with no hopping renormalization). Hence, it is not clear how valid this varational approach is, some benchmarking is really necessary.}
Unfortunately, we cannot follow the referee's line of argumentation here: How does not finding a feature in an inferior version of a method weaken the validity of a finding in the superior version? Our statement that lower variational degrees of freedom do not show a first-order transition is supposed to highlight that the physical mechanism behind this transition is the competition of kinetic and potential energy minimization, described in our variational approach by an abrupt change from a regime where mostly the effective interaction changes to one where mostly the effective hopping changes with $V$.
We have added a short discussion to the corresponding paragraph, clarifying the physical reason of why two parameter are crucial.
Related to this comment, we have showed in an earlier work (Phys. Rev. B 97, 165135 (2018)), that the variational principle with only one variational parameter does give a first-order transition, but this happens at much larger values of the non-local interaction $V$, where charge-density waves become relevant.
Concerning benchmarking of the variational method, we refer to our publication (Phys. Rev. B 94, 165141 (2016)) where we have performed promising benchmarks against dual boson calculations for the case of a single variational parameter on the square lattice. Additionally, our recent preprint (https://arxiv.org/abs/1901.11257) shows benchmarks of the method for the case of two variational parameters for exactly solvable systems.
\emph{1) The authors should say explicitly how the entropy calculations was done. }
We have included details of how we calculate the entropy to Appendix A on the free energy calculation.
\emph{2) In Fig. 4 and Fig. 5, the data are presented in $\tilde t$ and $\tilde U$ parameter space. It is really not clear how these numbers are related to real $U$ and $t$ values.}
In Fig. 4, the parameters of the original model (the real parameters $U$, $t$) are highlighted as blue lines in the space of variational parameters. This directly shows the influence nonlocal interactions have on the variational parameters. In Fig. 5, the parameters of the original model are those corresponding to the square markers, since $V=0$ means $\tilde U = U$ and $\tilde t = t$. For clarity, we have added the parameters of the original model in the caption of Fig. 5.
\emph{3) The authors claim that they find the band broadening effect of non-local interactions $V$. Their claim is based on some speculations of free energy behavior in $\tilde t$ parameter space. If the authors mean they see a similar effect as in Ref.~\url{https://journals.aps.org/prb/abstract/10.1103/PhysRevB.95.245130}, they should perhaps find a better way to support their claim.}
We actually do not mention band-widening in our manuscript. We did so on purpose although it may seem tempting to interpret a renormalization of the effective parameter $\tilde t$ as a band renormalization. First, however, $\tilde t$ only determines the band-width in the non-interacting (and weakly interacting) case. Second, the variational principle we use leads to variational parameters which optimize the free energy, i.e., ground-state properties, and not spectral properties.
\emph{4) Why in eq. 7, the original lattice dispersion function ϵk and not for the effective Hubbard model $\tilde \varepsilon_k$ is used.}
Eq. 7 of course contains the dispersion of the effective model. We have fixed our ambiguous notation in the revised manuscript.
\emph{5) Why the energy units (hopping t) are changes through the paper? In Fig. 6 the authors used $t=1.26$ and in Fig. 9, $t=1$ is used. This way your are studying the original extended Hubbard model for different bandwidth values.}
In our work, we decided to fix the energy units by the temperature ($T=0.1$) rather than a fixed hopping. Using the variational principle with the hopping $\tilde t$ as an effective parameter, it is quite natural to also consider a variable original hopping $t$.

---

## Round 3 · List of Changes

Added discussion in Sec. 3.1 on the significance of the discontinuities presented in Figs. 2 and 3 with respect to error estimates presented in Fig. 8.
Added the parameters of the presented original models to the caption of Fig. 5.
Added discussion in Sec. 3.3 on why two variational parameters are crucial for describing the first-order transition.
Clarified notation of Eq. 7 in Appendix A.
Extended appendix A on the calculation of the free energy with details of how we calculate the entropy (with a new Equation 8).
Added the parameters of the presented original models to the caption of Fig. 5.
Added discussion in Sec. 3.3 on why two variational parameters are crucial for describing the first-order transition.
Clarified notation of Eq. 7 in Appendix A.
Extended appendix A on the calculation of the free energy with details of how we calculate the entropy (with a new Equation 8).

---

## Editorial Decision

published